# GAP3D: Geometry-Aware Adaptive Planar Representations for 3D Occupancy Prediction

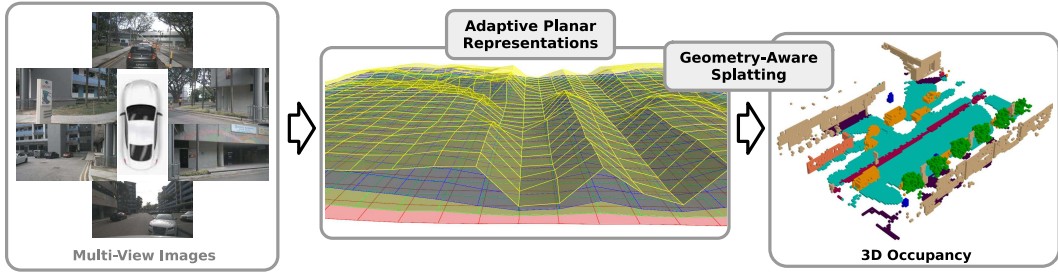

Figure 1: **Illustration of our GAP3D framework.** We introduce a new 3D representation, GAP3D, that constructs adaptive planar representations from multi-view images, where the height of each plane adapts dynamically to the underlying scene geometry. These planes are then fused into a global 3D occupancy volume via geometry-aware splatting, producing a dense and structured understanding of the environment.

## Abstract

We address the problem of 3D occupancy prediction from multi-view images, a task central to autonomous driving and embodied perception. Conventional methods typically employ fixed 3D voxel grids, which provide structured coverage of the entire scene but scale poorly in memory and computation as resolution increases. Sparse 3D representations, by contrast, improve efficiency by focusing only on occupied regions, but risk missing critical areas and often converge slowly. To overcome these limitations, we propose **Geometry-Aware Adaptive Planar Representations (GAP3D)**, a new adaptive 3D representation that combines the comprehensive coverage of voxel grids with the efficiency of sparse methods. GAP3D has two key components: (1) **Adaptive plane representations**, where planes sequentially partition vertical space via a stick-breaking process, concentrating representational capacity in regions with high occupancy likelihood; and (2) **Geometry-aware splatting**, which lifts the plane features into the full 3D occupancy volume through a differentiable Gaussian kernel, producing dense, spatially consistent predictions while preserving scene geometry. To further guide plane placement during early training, we introduce a height regularization loss that encourages alignment with scene structure. Experiments on the Occ3D-nuScenes benchmark demonstrate that GAP3D achieves the state-of-the-art performance while significantly reducing memory and computation compared to existing approaches.

## 1 Introduction

Understanding the full 3D structure of a scene is critical for autonomous systems that navigate, interact, or reason about complex real-world environments (Zhang et al., 2023a; Guo et al., 2025; Yao et al., 2018; Hoang et al., 2025; Han et al., 2025). 3D occupancy prediction provides a dense, voxelized representation that encodes both occupied and free space, enabling comprehensive reasoning over the environment, including regions not directly visible to sensors. To be effective in real-world applications, such representations should be (1) **complete**, providing global coverage of

the 3D scene to ensure that all regions are accounted for; (2) **compact**, since embedded devices often impose strict memory and computational constraints; and (3) **geometry-aware**, aligning the representational structure with the the geometry of the underlying scene to enhance accuracy and ensure geometrically coherent predictions.

To address the computational and memory challenges of 3D occupancy prediction, current methods primarily rely on either compact or sparse representations. Compact representations, such as Bird's-Eye-View (BEV) features or low-rank tensor decompositions (Yu et al., 2024; Zhao et al., 2024; Ma et al., 2024), achieve scalability by projecting or compressing the 3D volume into lower-dimensional forms. However, because much of the 3D scene consists of empty space, compressing the entire volume often wastes representational capacity on regions with little information, leaving fewer resources to capture occupied or geometrically important regions. Sparse methods reduce computation by predicting occupancy only at a limited set of points, exploiting the inherent sparsity of 3D scenes (Liu et al., 2025; Wang et al., 2024). While this selective prediction provides greater flexibility than fixed grids, the resulting points are irregular and unstructured, making it difficult to maintain global consistency and accurately represent scene geometry. More critically, because sampling is inherently selective, sparse methods are prone to missing small but important objects, which limits their reliability for downstream tasks.

In this work, we present Geometry-Aware Adaptive Planar Representations (GAP3D), a framework that combines the advantages of sparse and compact approaches: it adapts to the input by allocating more capacity to geometrically significant regions, while maintaining a structured organization along horizontal planes to ensure coverage across the entire scene. Specifically, GAP3D consists of two main components: (1) *Adaptive plane representations*, where the model predicts plane heights sequentially using a stick-breaking process, allowing vertical resolution to align with the density of the scene's geometry. Queries anchored to these planes extract features from the input images, capturing semantic and geometric cues relevant for occupancy prediction. By localizing feature extraction to adaptively placed planes, the model focuses capacity on informative regions while avoiding unnecessary computation over empty space. (2) *Geometry-aware splatting* fuses the plane features into a global 3D occupancy volume using a differentiable Gaussian kernel. This operation ensures that the voxel features correctly reflect the adaptive plane heights, preserving geometric structure along the vertical axis. The resulting voxel features are then decoded with a lightweight 3D CNN to predict the semantic occupancy of the scene. Fig. 1 shows an illustration of GAP3D.

Summary of contributions.

- We introduce GAP3D, a geometry-aware adaptive planar representation for 3D semantic occupancy prediction. Instead of a fixed 3D grid or irregular point samples, GAP3D uses a set of 2D planes whose locations are conditioned on the input images, yet remain structured in the $xy$-axes. This yields global coverage while concentrating capacity along geometrically informative regions.

- We introduce an adaptive mechanism for predicting plane heights based on a stick-breaking process, which partitions the vertical extent of the scene into a sequence of non-overlapping segments. This formulation guarantees several desirable properties: the predicted plane heights are always monotonically increasing, since each new plane is defined as a fraction of the remaining vertical range, and they remain bounded within the scene height. Together, these properties offer flexibility by allowing the density of planes to vary across locations, adapting the vertical resolution to the underlying scene geometry.

- We fuse plane-anchored features into a voxel volume using a differentiable Gaussian kernel that weighs contributions by the geometric distance to the predicted plane height at each $(x, y)$ location. This respects the learned geometry when mapping from planes to voxels, promoting geometric consistency between adaptive heights and fused 3D features.

- GAP3D achieves the state-of-the-art performance on the Occ3D-nuScenes benchmark with a RayIoU score of 42.8%, while requiring moderately less computation than previous methods. This establishes a desirable trade-off between accuracy and efficiency, demonstrating that adaptive planar representations can outperform both compressed and sparse baselines in real-world 3D occupancy prediction.

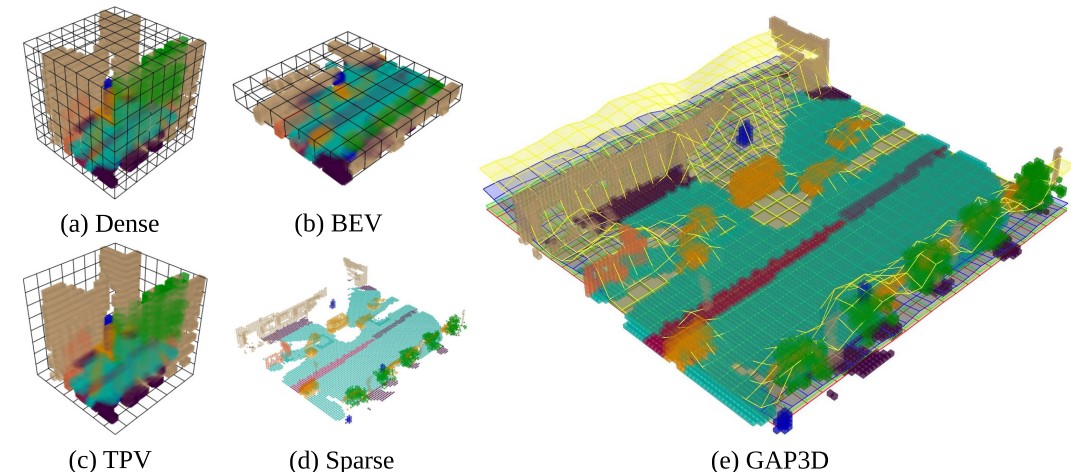

(a) Dense  (b) BEV

(c) TPV  (d) Sparse  (e) GAP3D

Figure 2: Comparison of 3D occupancy representations. Dense representations use a 3D voxel grid with uniform resolution, capturing all regions but at high computational cost. BEV representations compresses the 3D volume into a bird's-eye-view (BEV) plane, reducing computation but losing vertical detail. TPV represents the 3D scene using three orthogonal planes, which is computationally efficient but prone to inaccuracies due to its implicit representation. Sparse methods aim to predict occupancy only at selected points, saving computation but leaving many regions unobserved. GAP3D uses adaptive planar representations, concentrating representational capacity on geometrically significant regions while maintaining global coverage and structural consistency.

## 2 RELATED WORK

### 2.1 DENSE 3D REPRESENTATIONS

Dense methods predict occupancy over fixed volumetric grids, ensuring full coverage of the 3D scene but at the cost of high memory and computation. Early approaches extended dense 2D BEV pipelines into 3D voxel grids (Cao & De Charette, 2022; Wei et al., 2023; Zhang et al., 2023b; Tian et al., 2024), typically by lifting multi-view image features into the voxel space. To improve scalability, recent works introduced compact alternatives that compress 3D information into lower-dimensional forms, such as bird's-eye-view projections (Li et al., 2023b;a;c), or low-rank tensor decompositions (Zhao et al., 2024; Ma et al., 2024). While these approaches alleviate some of the computational burden, they still waste capacity on large empty regions, and the compressed forms limit the ability to capture fine geometric details. As illustrated in Fig. 2, GAP3D instead preserves global coverage while avoiding wasted computation by organizing adaptive planes that concentrate capacity where geometry is present.

### 2.2 SPARSE REPRESENTATIONS

To overcome the inefficiency of dense grids, sparse methods predict occupancy only at selected points, leveraging the fact that most voxels are empty. SparseOcc (Liu et al., 2025) directly learns sparse voxel features, focusing computation on occupied regions, while OPUS (Wang et al., 2024) reformulates occupancy as a set prediction task in which query points are iteratively refined to capture geometry. These approaches are more efficient, but their reliance on irregular sampling often leads to missed objects in occluded or fine-grained regions, and the lack of global structure makes it difficult to maintain scene-level consistency. GAP3D addresses these issues by aligning adaptive planes with scene geometry, ensuring structural consistency while retaining efficiency.

## 3 METHOD

In this section, we introduce Geometry-Aware Adaptive Planar Representations (GAP3D), a structured 3D occupancy prediction framework based on sequential plane representations. Adaptive plane

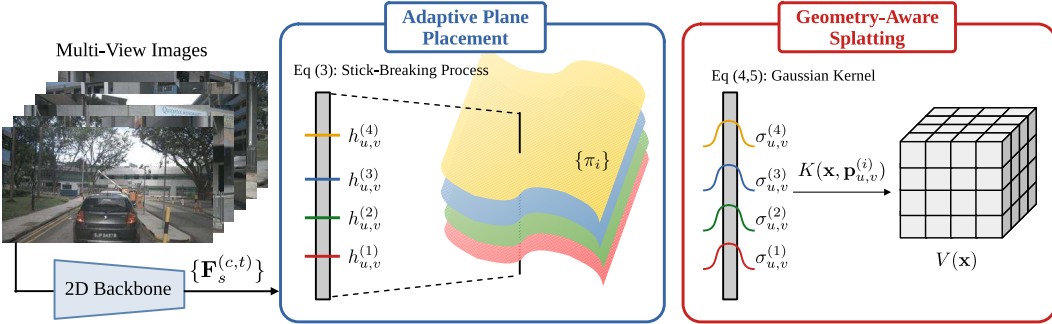

Figure 3: **Visualization of framework.** GAP3D predicts adaptive planes at varying heights across the scene using a stick-breaking process (Sec. 3.1), aligning them with geometry-rich regions. These plane features are then projected into the 3D occupancy volume via geometry-aware splatting with Gaussian kernels (Sec. 3.2), producing spatially consistent voxel features.

placement (Sec. 3.1) allocates resolution to geometry-rich regions, alleviating the inefficiency of uniform voxel discretization. Geometry-aware splatting (Sec. 3.2) projects the refined plane features into the global 3D occupancy volume via a differentiable Gaussian kernel, preserving gradients while maintaining spatial structure. By coupling adaptive placement with splatting, GAP3D encourages planes to concentrate in geometrically significant regions, enabling a compact and interpretable 3D representation that focuses computation where the scene structure is most informative. Fig. 3 provides a visualization of our proposed framework.

### 3.1 ADAPTIVE PLANE PLACEMENT

3D occupancy representations are typically based on 3D voxel grids, which provide a structured representation of the scene, but are computationally expensive and inefficient since most voxels correspond to empty space (*e.g.*, $200 \times 200 \times 16$ for Occ3D-nuScenes, around $90\%$ empty voxels). Sparse representations (Liu et al., 2025; Wang et al., 2024) focus on occupied regions, improving efficiency, but their irregular sampling risks missing small or thin structures and can lead to discontinuities. Our framework offers a natural compromise: instead of committing to either dense voxels or fully unstructured sampling, we represent the scene with a sequence of *adaptively placed planes*. This combines the efficiency and regularity of voxel grids with the flexibility of adaptive placement, allowing the representation to concentrate resolution near geometrically significant regions. By progressively placing planes from the bottom up, the model naturally aligns with real-world scenes, where geometry typically starts at the floor and grows upward, resulting in a representation that is both compact and physically grounded. Fig. 2 shows a comparison with existing 3D occupancy representations.

We represent the 3D scene using a sequence of adaptive BEV planes, where each plane acts as a structured layer of queries covering the horizontal extent of the scene. Formally, each plane $\pi_i$ is a 2D grid of queries:

$$\pi_i = \{q_{u,v}^{(i)} \mid (u,v) \in \Omega\}, \quad q_{u,v}^{(i)} = (\mathbf{p}_{u,v}^{(i)}, \mathbf{e}_{u,v}^{(i)}), \tag{1}$$

where $(u,v)$ indexes the grid cell in the BEV plane, and $\Omega = \{0, \dots, U-1\} \times \{0, \dots, V-1\}$ denotes the discrete set of horizontal grid coordinates with resolution $U \times V$. Each query $q_{u,v}^{(i)}$ consists of a 3D anchor position $\mathbf{p}_{u,v}^{(i)} = (x_u, y_v, h_{u,v}^{(i)})$ and a feature embedding $\mathbf{e}_{u,v}^{(i)} \in \mathbb{R}^d$ that encodes local scene information. The height $h_{u,v}^{(i)}$ is adaptively predicted for each query, allowing planes to focus resolution where the geometry is most significant. Conceptually, each query acts as a probe that summarizes occupancy and features at a specific location, enabling the plane to compress a horizontal slice of the 3D scene into a 2D feature map. By arranging queries in planes instead of scattered points, the model preserves *spatial regularity* for efficient processing while maintaining flexibility to adapt to varying scene structures.

**Adaptive Placement via Stick-Breaking Process.** To effectively allocate vertical resolution, the plane heights must satisfy two key properties: (a) the last plane should not exceed the vertical extent of the prediction space, and (b) the planes should concentrate around geometrically significant regions along each column. To achieve these goals, we employ a *stick-breaking process*, which is well-suited for sequential, adaptive allocation of a bounded range. By treating the remaining vertical space as a "stick", each plane can claim a fraction proportional to its predicted importance, ensuring that cumulative heights stay within bounds while placing more space planes near regions with dense geometry. This approach naturally enforces both constraints: the total height never exceeds the scene extent, and the distribution of planes adapts to the input geometry.

Specifically, for each plane $\pi_i$, the height ratio for a query at $(u, v)$ is predicted from the previous plane's feature:

$$r_{u,v}^{(i)} = \sigma(\text{MLP}(\mathbf{e}_{u,v}^{(i-1)})) \in (0, 1), \tag{2}$$

where MLP is a linear layer and $\sigma$ denotes the sigmoid function. The absolute normalized height $\bar{h}_{u,v}^{(i)}$ is then computed as:

$$\Delta \bar{h}_{u,v}^{(i)} = r_{u,v}^{(i)} \cdot \left(1 - \sum_{k<i} \Delta \bar{h}_{u,v}^{(k)}\right), \quad \bar{h}_{u,v}^{(i)} = \sum_{k \leq i} \Delta \bar{h}_{u,v}^{(k)}. \tag{3}$$

Here, $\Delta \bar{h}_{u,v}^{(i)}$ represents the height increment for the current plane, and the cumulative sum $\bar{h}_{u,v}^{(i)}$ gives the absolute vertical position. In practice, we implement a numerically stable variant in log-space to avoid exploding gradients (see Sec. A.2). By allocating the vertical range in this way, each plane naturally focuses on geometrically significant regions, ensuring efficient and adaptive coverage of the 3D scene.

## 3.2 Geometry-Aware Splatting

To reconstruct the full 3D occupancy volume from adaptive plane features, a naive channel-to-height mapping (Yu et al., 2024) is insufficient. These approaches treat features independently of their 3D anchor positions, allowing the network to map features to heights without regard for the actual geometry. As a result, the vertical structure can be misaligned with the scene, reducing geometric consistency. Instead, we employ *geometry-aware splatting*, which projects each plane's queries into the 3D volume using a differentiable Gaussian kernel. This operation preserves the spatial relationship of features, ensures smooth assignment across voxels, and allows end-to-end learning.

Formally, for a plane $\pi_i$ with queries $q_{u,v}^{(i)}$, the voxel feature at location $\mathbf{x} = (x, y, z)$ is computed as:

$$V(\mathbf{x}) = \sum_{(u,v) \in \Omega} K(\mathbf{x}, \mathbf{p}_{u,v}^{(i)}) \cdot \mathbf{e}_{u,v}^{(i)}, \tag{4}$$

where $K(\mathbf{x}, \mathbf{p})$ is a differentiable Gaussian kernel:

$$K(\mathbf{x}, \mathbf{p}_{u,v}^{(i)}) = \exp\left(-\frac{\|\mathbf{x} - \mathbf{p}_{u,v}^{(i)}\|_2^2}{2\sigma_{u,v}^{(i)2}}\right). \tag{5}$$

Here, $\sigma_{u,v}^{(i)}$ controls the spatial spread of query $q_{u,v}^{(i)}$. Similar to the height ratio in adaptive placement, $\sigma_{u,v}^{(i)}$ is predicted by an MLP from the query embedding $\mathbf{e}_{u,v}^{(i)}$. This allows each query to dynamically adapt its influence radius according to local scene geometry, leading to smoother splatting in large open regions and sharper boundaries around fine structures.

**Discussion.** While geometry-aware splatting preserves spatial structure and ensures smooth assignment of plane features into the 3D volume, this operation alone does not explicitly guide the adaptive plane heights to align with scene geometry. This is because the plane features encode both occupied and empty regions, so gradient signals from splatting are distributed across all voxels rather than concentrated on geometrically significant locations. To address this, in the next subsection we introduce a *height regularization loss* during early training, which encourages planes to place queries closer to regions with meaningful geometry and reinforces the overall adaptive placement strategy.

## 3.3 GAP3D ARCHITECTURE AND LOSS FUNCTION

**Architecture Overview.** Our framework builds upon a 2D image encoder backbone (e.g., ResNet-50 or Swin Transformer) to extract multi-scale, multi-view image features $\mathbf{F}_s^{(c,t)}$, where $\mathbf{F}_s^{(c,t)} \in \mathbb{R}^{C_s \times H_s \times W_s}$ denotes the feature map at scale $s$, camera $c$, and timestamp $t$. The GAP3D representation consists of a hierarchy of BEV grids $\{\mathcal{G}_\ell\}_{\ell=1}^L$, where each grid $\mathcal{G}_\ell$ consists of $K$ adaptive planes $\{\pi_{\ell,i}\}_{i=1}^K$. Queries within each plane interact with the multi-scale, multi-view image features $\mathbf{F}_s^{(c,t)}$ to update their embeddings. Planes are then projected into the 3D voxel volume via geometry-aware splatting, producing voxel features $V^{(\ell)}$. Features from all BEV levels are fused in a coarse-to-fine refinement scheme: lower-resolution grids capture global context, while higher-resolution grids refine geometric details. The fused 3D volume is decoded by a lightweight 3D CNN head to produce dense occupancy predictions.

**Image Sampling and Adaptive Mixing.** Each query in the adaptive planes aggregates information by sampling multiple points from the multi-scale, multi-view image features extracted by the backbone. The sampled points are projected from the 3D query positions into all camera views and timestamps, capturing rich spatial and temporal context. These point-level features are then combined with the query embeddings through and adaptive mixing mechanism, allowing the model to emphasize informative views while down-weighting occluded or less-relevant regions. This process enriches the plane representations before they are projected into the 3D voxel volume via geometry-aware splatting. Please refer to Sec. A.3 and Sec. A.4 for detailed implementation of the sampling and mixing operations.

**Modeling Long-Range Dependencies.** Accurate occupancy prediction requires reasoning over spatially distant regions and maintaining geometric consistency throughout the scene. To achieve this, we employ selective state-space models (Gu & Dao, 2023; Liu et al., 2024) for efficient global context modeling. Within each adaptive plane, 2D Mamba captures long-range interactions among queries in the horizontal BEV domain. To further strengthen vertical consistency, we introduce 3D Mamba before splatting, where the sequence order extends the 2D scan operations by including neighboring queries across planes along the $z$-axis. Each query is further augmented with a 3D positional embedding derived from its anchor location $(x_u, y_v, h_{u,v}^{(i)})$, so that the model explicitly encodes the adaptive height position when reasoning over sequences. This enables queries to share information both laterally and vertically while still in the adaptive representation, ensuring that the subsequent splatting step receives geometry-aware, globally contextualized features.

**Loss Functions** We follow the voxel-wise training loss functions used by Li et al. (2023c). These consist of the distance-aware focal loss $\mathcal{L}_{\text{focal}}$ proposed by Xie et al. (2022), the geometric and semantic affinity losses ($\mathcal{L}_{\text{scal}}^{\text{geo}}$ and $\mathcal{L}_{\text{scal}}^{\text{sem}}$) from MonoScene (Cao & De Charette, 2022), and the Lovász-softmax $\mathcal{L}_{\text{lov}}$ (Berman et al., 2018).

To guide the adaptive plane placement during early training, we introduce a height regularization loss that applies only to the lowest and highest planes along each BEV grid and only if occupied voxels are present:

$$\mathcal{L}_{\text{height}} = \frac{1}{|\Omega|} \sum_{(u,v) \in \Omega} \mathbf{1}\{(u,v) \in \mathcal{O}\} \left[ \ell_{\text{reg}}(h_{u,v}^{(1)}, h_{\min,u,v}^*) + \ell_{\text{reg}}(h_{u,v}^{(K)}, h_{\max,u,v}^*) \right], \quad (6)$$

where $h_{u,v}^{(1)}$ and $h_{u,v}^{(K)}$ are the predicted heights of the first and last planes at BEV cell $(u,v)$, $h_{\min,u,v}^*$ and $h_{\max,u,v}^*$ are the corresponding reference heights from the ground-truth occupnacy, and $\ell_{\text{reg}}$ is a distance metric such as the $\ell_1$ or Huber loss. The weight of the height regularization term, $\lambda_{\text{height}}$, is annealed during training using a cosine schedule, starting with higher values in early epochs to encourage alignment with geometrically significant regions, and gradually decreases to zero as training progresses, allowing the model more flexibility in adaptive plane placement.

The total training loss is a weighted combination of all objectives:

$$\mathcal{L}_{\text{total}} = \lambda_{\text{focal}}\mathcal{L}_{\text{focal}} + \lambda_{\text{geo}}\mathcal{L}_{\text{scal}}^{\text{geo}} + \lambda_{\text{sem}}\mathcal{L}_{\text{scal}}^{\text{sem}} + \lambda_{\text{lov}}\mathcal{L}_{\text{lov}} + \lambda_{\text{height}}\mathcal{L}_{\text{height}}. \quad (7)$$

This combination ensures both accurate voxel-wise occupancy prediction and meaningful adaptive plane placement.

Table 1: RayIoU Liu et al. (2025) results on Occ3D-nuScenes Tian et al. (2024). (8f) and (16f) denote the number of frames used to predict the semantic occupancy of the current frame. FPS is measured using a single A100 GPU with PyTorch FP32 backend for all experiments. We report results shown in either Yu et al. (2024); Wang et al. (2024) or their respective papers.

| Method | Backbone | Input Size | RayIoU(%)↑ | RayIoU$_{1m}$(%)↑ | RayIoU$_{2m}$(%)↑ | RayIoU$_{4m}$(%)↑ | FPS |
|---|---|---|---|---|---|---|---|
| RenderOcc (Pan et al., 2024) | Swin-B | $1408 \times 512$ | 19.5 | 13.4 | 19.6 | 25.5 | - |
| BEVFormer (Li et al., 2022) | R101 | $1600 \times 900$ | 32.4 | 26.1 | 32.9 | 38.0 | 3.0 |
| BEVDet-Occ (Huang et al., 2021) | R50 | $704 \times 256$ | 29.6 | 23.6 | 30.0 | 35.1 | 2.6 |
| BEVDet-Occ (8f) (Huang et al., 2021) | R50 | $704 \times 384$ | 32.6 | 26.6 | 33.1 | 38.2 | 0.8 |
| FB-Occ (16f) (Li et al., 2023c) | R50 | $704 \times 256$ | 33.5 | 26.7 | 34.1 | 39.7 | 10.3 |
| SparseOcc (8f) (Liu et al., 2025) | R50 | $704 \times 256$ | 34.0 | 28.0 | 34.7 | 39.4 | 17.3 |
| SparseOcc (16f) (Liu et al., 2025) | R50 | $704 \times 256$ | 35.1 | 29.1 | 35.8 | 40.3 | 12.5 |
| Panoptic-FlashOcc (8f) (Yu et al., 2024) | R50 | $704 \times 256$ | 38.5 | 32.8 | 39.3 | 43.4 | **35.6** |
| OPUS-L (8f) (Wang et al., 2024) | R50 | $704 \times 256$ | 41.2 | 34.7 | 42.1 | 46.7 | 7.2 |
| STCOcc (16f) (Liao et al., 2025) | R50 | $704 \times 256$ | 41.7 | 36.2 | 42.7 | 46.4 | 4.4 |
| STCOcc (16f) (Liao et al., 2025) | R50 | $1408 \times 512$ | 42.1 | 36.9 | 42.8 | 46.7 | - |
| GAP3D (8f) | R50 | $704 \times 256$ | **42.8** | **37.8** | **43.6** | **47.0** | 9.0 |

# 4 EXPERIMENTS

In this section, we first describe the experimental setup used in Sec. 4.1, followed by a brief summary of results compared to other state-of-the-art methods in Sec. 4.2. Finally, we conduct an ablation study to analyze the contribution of each component, as shown in Sec. 4.3.

## 4.1 EXPERIMENTAL SETUP

**Dataset.** We conduct experiments on the Occ3D-nuScenes benchmark (Tian et al., 2024), a large-scale dataset for semantic occupancy prediction built on top of nuScenes. The dataset provides 3D voxel annotations with $[200, 200, 16]$ resolution along $(x, y, z)$ axes, spanning a spatial range of $[-40, 40]$ m in the horizontal plane and $[-1.0, 5.4]$ m vertically. Each voxel corresponds to one of 18 semantic categories, including an empty class. Following standard practice, we use the official train/val/test splits with 700, 150, and 150 scenes, respectively. Since full 3D supervision is challenging to annotate, the benchmark additionally provides visibility masks indicating occluded regions.

**Evaluation metric.** The mean Intersection-over-Union (mIoU) has been the standard metric for evaluating semantic occupancy on Occ3D-nuScenes (Tian et al., 2024), but because it is computed under camera visibility masks, it mainly reflects accuracy in visible regions and fails to capture a model's ability to complete occluded structures. To better assess reconstruction quality, we adopt RayIoU as our primary metric (Liu et al., 2025). RayIoU traces camera rays from multiple view-points and compares predictions along each ray at different depth thresholds (1 m, 2 m, and 4 m). This design penalizes false positives in occluded areas and provides a more reliable measure of 3D scene completeness.

**Implementation details.** We adopt ResNet-50 as the image backbone with input resolution $704 \times 256$, following standard practice to ensure comparability with prior works. For the 3D representation, we use four adaptive planes with latent channel dimension 80, and BEV feature resolutions of $100 \times 100$, $50 \times 50$, and $25 \times 25$ across scales. Models are trained on 8 NVIDIA A100 GPUs for 60 epochs in total: a global batch size of 8 is used for the first 24 epochs, which is then increased to 32 for the remaining 36 epochs. Optimization is performed with AdamW using a learning rate of $1e-4$ and weight decay of $0.01$. A regularization loss is applied during the first 6 epochs, with a weighting coefficient $\lambda_{\text{height}}$ initialized at 1 and linearly annealed to 0. For data augmentation, we follow the same image- and BEV-level strategies as proposed in Li et al. (2023c).

## 4.2 MAIN RESULTS

**Quantitative results.** Tab. 1 reports RayIoU on the Occ3D-nuScenes benchmark. GAP3D attains the highest overall RayIoU of 42.8%, outperforming the previous best STCOcc (Liao et al., 2025) despite using fewer frames (8f vs. 16f) and a lower input resolution. GAP3D is consistently

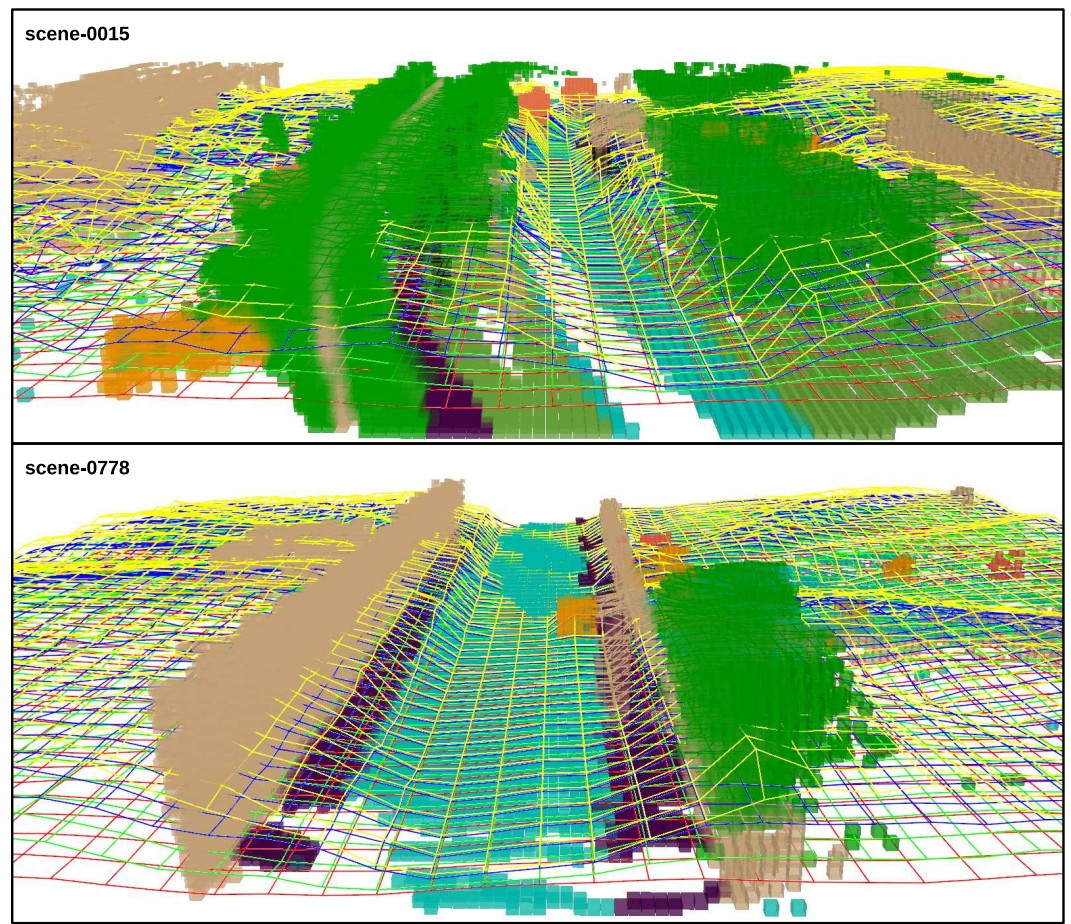

scene-0015

scene-0778

Figure 4: **Visualization of adaptive planes predicted by GAP3D.** Front-view examples of adaptive planes are shown, with colors mapped from bottom to top in the order: red, green, blue, and yellow.

stronger at all distance thresholds: 37.8% at 1 m, 43.6% at 2 m, and 47.0% at 4 m, indicating robust improvements at near- and far-range predictions.

In addition to accuracy, GAP3D maintains competitive runtime. It operates at 9.0 FPS on a single A100 GPU, which is considerably faster than OPUS-L (7.2 FPS) and estimated to be at least 2.0× faster than STCOcc. Compared to SparseOcc, GAP3D demonstrates clear gains: with only 8 frames, GAP3D exceeds SparseOcc (16f) by +7.7 RayIoU (42.8% vs. 35.1%) while sacrificing only a small amount of speed (9.0 FPS vs 12.5 FPS). Overall, these results establish GAP3D as the new state-of-the-art on Occ3D-nuScenes, offering a strong accuracy–efficiency trade-off compared to both dense voxel and sparse query formulations.

**Qualitative results.** To better illustrate how GAP3D allocates representational capacity, Fig. 4 visualizes the predictive adaptive planes as grids placed at different heights in the scene. The model consistently positions planes at semantically meaningful locations, showing elevations around vehicles, trees, and building structures. This behavior shows GAP3D adapts its resolution according to scene structure: concentrating capacity where geometry is dense and reducing it in empty or less informative regions. The resulting allocation provides a compact yet geometrically aware representation of the 3D environment.

Table 2: Ablation on proposed components.

| Design | RayIoU (%) ↑ |
|---|---|
| Fixed planes (baseline) | 38.2 |
| Adaptive planes (naive lifting) | 38.9 |
| Adaptive planes + splatting (ours) | **39.3** |

Table 3: Ablation on number of planes.

| # Planes | RayIoU (%) ↑ | FPS ↑ |
|---|---|---|
| 2 | 37.3 | 11.3 |
| 4 | 38.7 | 8.1 |
| 6 | 38.4 | 6.3 |
| 8 | 38.9 | 4.4 |

## 4.3 ABLATION STUDY

Table 2 summarizes a controlled ablation isolating the contributions of adaptive plane placement and geometry-aware splatting. All variants share the same backbone, network structure, and hyper-parameters, and are trained for 12 epochs to ensure a fair comparison. Starting from the fixed-plane baseline at 38.2 RayIoU, replacing uniform planes with adaptive placement improves performance to 38.9 (+0.7), showing that allocating vertical resolution to geometry-dense regions yields more accurate predictions. Adding geometry-aware splatting further increases accuracy to 39.3 (+0.4), as the Gaussian kernel projection enforces spatially coherent lifting of plane features into the voxel volume. Overall, these results demonstrate a cumulative gain of +1.1 RayIoU, confirming that both adaptive placement and splatting contribute complementary improvements.

We also study the effect of varying the number of adaptive planes, shown in Table 3. We reduce the latent channel dimension to 64, due to computational costs as the number of planes increase. Using only 2 planes reduces accuracy to 37.3 RayIoU, though it offers the fastest runtime at 11.3 FPS. Increasing to 4 planes provides the best balance, boosting accuracy to 38.7 (+1.4) while maintaining competitive efficiency at 8.1 FPS. Interestingly, adding more planes does not consistently improve performance: with 6 planes, accuracy slightly decreases 38.4 despite slower inference at 6.3 FPS. At 8 planes, accuracy increases slightly to 38.9 but speed further drops to 4.4 FPS. These results indicate that simply adding more planes is not always advantageous, particularly on the Occ3D-nuScenes benchmark where the vertical voxel resolution is 16, constraining the benefit of finer partitioning. In practice, 4 planes offer the most effective trade-off between accuracy and efficiency, and we adopt this setting for all main experiments.

## 5 CONCLUSION

We have presented GAP3D, a novel framework for 3D semantic occupancy prediction that leverages structured, adaptive planar representations combined with geometry-aware splatting. By predicting plane heights conditioned on the input scene, GAP3D concentrates representational capacity on geometrically significant regions while avoiding wasted computation over empty space. Geometry-aware splatting fuses these planes into a coherent 3D occupancy volume, preserving vertical detail and maintaining global consistency. Extensive experiments on the Occ3D-nuScenes benchmark show that GAP3D achieves state-of-the-art RayIoU performance while maintaining moderate computational cost, outperforming prior methods such as STCOcc and OPUS, and ablation studies confirm the importance of adaptive plane placement and geometry-aware splatting.

Despite these strengths, GAP3D's plane-based design may be less effective in scenarios with more vertically balanced voxel grids, such as indoor scenes, where fine vertical structures are relatively more prominent. Future work could investigate ways to adapt planar representations for such environments, incorporate richer priors about vertical geometry, or combine GAP3D with complementary 3D representations to further improve accuracy in vertically complex scenes.

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

# A APPENDIX

## A.1 USE OF LARGE LANGUAGE MODELS (LLMS)

In line with the ICLR 2026 policies on large language model (LLM) usage, we disclose that an LLM was employed only to polish the language of this manuscript. Its role was strictly limited to improving grammar, clarity, and readability of text already drafted by the authors. The LLM was not involved in research ideation, experiment design, or the conceptual framing of the paper. We have independently verified all claims, results, and interpretations, and take full responsibility for the content of this work.

## A.2 STICK-BREAKING PROCESS IN LOG SPACE

For each BEV cell $(u, v)$, the vertical allocation ratios $r_{u,v}^{(i)}$ are obtained through a stick-breaking process. We first obtain *unnormalized logits* from the previous plane's query embedding via an MLP:

$$\alpha_{u,v}^{(i)} = \text{MLP}(\mathbf{e}_{u,v}^{(i-1)}). \tag{8}$$

Given these logits, the stick-breaking ratios are

$$r_{u,v}^{(1)} = \sigma(\alpha_{u,v}^{(1)}), \quad r_{u,v}^{(i)} = \sigma(\alpha_{u,v}^{(i)}) \prod_{j=1}^{i-1}(1 - \sigma(\alpha_{u,v}^{(j)})), \quad i = 2, \ldots, K, \tag{9}$$

where $\sigma(x) = \frac{1}{1+e^{-x}}$ is the sigmoid function. Each ratio $r_{u,v}^{(i)}$ specifies the fraction of the remaining vertical range allocated to plane $\pi_i$.

**Numerical stability.** Directly evaluating the product form above can lead to numerical overflow when many factors are multiplied. To address this, we compute the stick-breaking process in *log space*:

$$\log r_{u,v}^{(1)} = \log \sigma(\alpha_{u,v}^{(1)}), \quad \log r_{u,v}^{(i)} = \log \sigma(\alpha_{u,v}^{(i)}) + \sum_{j=1}^{i-1} \log(1 - \sigma(\alpha_{u,v}^{(j)})). \tag{10}$$

Note the identities

$$\log \sigma(x) = -\text{softplus}(-x), \tag{11}$$
$$\log(1 - \sigma(x)) = -\text{softplus}(x), \tag{12}$$

with $\text{softplus}(x) = \log(1 + e^x)$. Substituting gives the stable log-space form:

$$\log r_{u,v}^{(1)} = -\text{softplus}(-\alpha_{u,v}^{(1)}), \tag{13}$$

$$\log r_{u,v}^{(i)} = -\text{softplus}(-\alpha_{u,v}^{(i)}) - \sum_{j=1}^{i-1} \text{softplus}(\alpha_{u,v}^{(j)}), \quad i = 2, \ldots, K. \tag{14}$$

## A.3 SAMPLING FROM IMAGE FEATURES

We describe the procedure for sampling multi-view image features into query embeddings. This follows the formulation of SparseOcc (Liu et al., 2025), but is adapted to our adaptive plane representation.

**Projection with sampling offsets.** Each query anchor is denoted as $\mathbf{p}_{u,v}^{(i)} = (x_u, y_v, h_{u,v}^{(i)})$. We first generate a set of *sampling offsets* from the query embedding. Specifically, a linear layer predicts $P$ offsets per query:

$$\Delta \mathbf{p}_{u,v,k}^{(i)} = \text{Linear}(\mathbf{e}_{u,v}^{(i)}), \quad k = 1, \ldots, P, \tag{15}$$

where each $\Delta \mathbf{p} \in \mathbb{R}^3$ is expressed in normalized coordinates. The effective sampling locations are

$$\hat{\mathbf{p}}_{u,v,k}^{(i)} = \mathbf{p}_{u,v}^{(i)} + \Delta \mathbf{p}_{u,v,k}^{(i)}. \tag{16}$$

Each $\hat{\mathbf{p}}_{u,v,k}^{(i)}$ is then projected into the image plane of camera $c$ at frame $t$:

$$\tilde{\mathbf{x}}_{u,v,c,t,k}^{(i)} = \mathrm{proj}(K_c, T_c, \hat{\mathbf{p}}_{u,v,k}^{(i)}), \tag{17}$$

where $K_c$ and $T_c$ denote camera intrinsics and extrinsics.

**Multi-scale sampling.** For each backbone feature map $\mathbf{F}_s^{(c,t)} \in \mathbb{R}^{C_s \times H_s \times W_s}$, bilinear interpolation is applied directly:

$$\mathbf{f}_{u,v,c,t,s,k}^{(i)} = \mathrm{bilinear}\left(\mathbf{F}_s^{(c,t)}, \tilde{\mathbf{x}}_{u,v,c,t,k}^{(i)}\right). \tag{18}$$

In addition to offsets, the query embedding predicts a set of scale weights $\beta_{u,v,s}^{(i)}$ via another linear layer. We normalize them with a softmax across scales:

$$w_{u,v,s}^{(i)} = \frac{\exp(\beta_{u,v,s}^{(i)})}{\sum_{s'} \exp(\beta_{u,v,s'}^{(i)})}. \tag{19}$$

The final sampled feature for query $(i, u, v)$ is then a weighted sum over scales:

$$\mathbf{f}_{u,v,c,t,k}^{(i)} = \sum_s w_{u,v,s}^{(i)} \mathbf{f}_{u,v,c,t,s,k}^{(i)}. \tag{20}$$

## A.4 ADAPTIVE MIXING

After sampling, each query $q_{u,v}^{(i)}$ is associated with a set of features $\{\mathbf{f}_{u,v,c,t,k}^{(i)}\}$ from multiple cameras $c$, frames $t$, and offsets $k$, already aggregated across scales with weights $w_{u,v,s}^{(i)}$ as described in Appendix A.3. The goal of adaptive mixing is to fuse these features into a single query embedding.

**Channel mixing.** We organize the sampled features into a matrix $\mathbf{F}_{u,v}^{(i)} \in \mathbb{R}^{N \times C}$, where $N = C_{\mathrm{cam}} \times T \times P$ is the number of valid samples for the query and $C$ is the channel dimension. From the query embedding $\mathbf{e}_{u,v}^{(i)}$, a linear layer predicts dynamic channel weights:

$$W_c = \mathrm{Linear}(\mathbf{e}_{u,v}^{(i)}) \in \mathbb{R}^{C \times C}. \tag{21}$$

Channel mixing is then applied as

$$M_c(\mathbf{F}_{u,v}^{(i)}) = \mathrm{ReLU}(\mathrm{LN}(\mathbf{F}_{u,v}^{(i)} W_c)). \tag{22}$$

**Point mixing.** We next transpose the feature matrix to $\mathbf{F}_{u,v}^{(i)\top} \in \mathbb{R}^{C \times N}$, and use another linear layer to predict dynamic point weights:

$$W_p = \mathrm{Linear}(\mathbf{e}_{u,v}^{(i)}) \in \mathbb{R}^{N \times N}. \tag{23}$$

Point mixing is applied as

$$M_p(\mathbf{F}_{u,v}^{(i)}) = \mathrm{ReLU}(\mathrm{LN}(\mathbf{F}_{u,v}^{(i)\top} W_p)). \tag{24}$$

**Aggregation.** The outputs from channel and point mixing are flattened and aggregated by a linear layer, yielding

$$\tilde{\mathbf{e}}_{u,v}^{(i)} = \mathrm{Linear}(M_p(M_c(\mathbf{F}_{u,v}^{(i)}))). \tag{25}$$

The final query embedding is updated with a residual connection:

$$\mathbf{e}_{u,v}^{(i)} \leftarrow \mathbf{e}_{u,v}^{(i)} + \mathrm{LN}(\tilde{\mathbf{e}}_{u,v}^{(i)}). \tag{26}$$

## A.5 ADDITIONAL QUALITATIVE RESULTS

We provide additional qualitative comparisons of OPUS-L (Wang et al., 2024) and GAP3D. Grey voxels indicate regions ignored during mIoU evaluation on Occ3D-nuScenes. Camera-visibility masks corresponding to empty annotations are removed for clarity.

(a) Ground-truth    (b) OPUS-L    (c) GAP3D

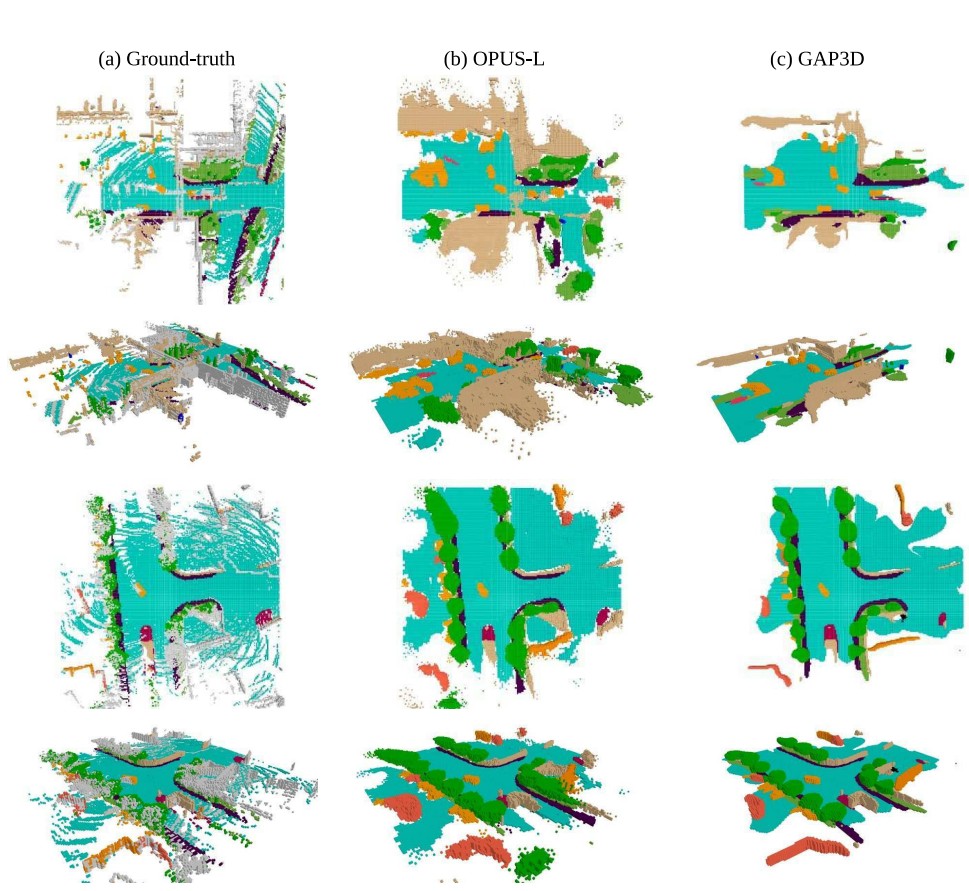

Figure 5: **Visualization of predicted voxels for OPUS-L and GAP3D on Occ3D-nuScenes.**

