# OpenReview forum: "GAP3D: Geometry-Aware Adaptive Planar Representations for 3D Occupancy Prediction"
_ICLR.cc/2026/Conference — Submitted to ICLR 2026_

### Official Review · Reviewer_uiDp · 2025-10-26

**Soundness:** 2
**Presentation:** 3
**Contribution:** 2
**Rating:** 4
**Confidence:** 5

**Summary:**

The paper proposes a geometry-aware adaptive planar representation (GAP3D) to combine the advantages of grid-based representations and sparse representations. GAP3D first adaptively places planes along the vertical dimension with a stick-breaking formulation, and then converts the planes to 3D occupancy with the geometry-aware splatting technique. Experiments show that GAP3D achieves state-of-the-art performance on Occ3D-nuScenes.

**Strengths:**

1. The motivation to combine the global coverage of grid-based representations and the efficiency of sparse representations is reasonable.
2. The performance of GAP3D on Occ3D-nuScenes is good compared with other methods.
3. The paper is well-written and easy to follow.

**Weaknesses:**

1. Lack of related work. The paper does not cite closely related work such as TPVFormer, GaussianFormer, etc.
2. Lack of novelty. The proposed GAP3D can be regarded as constraining the positions of Gaussian primitives in GaussianFormer to grid centers in the xy plane, and the paper fails to explain why such a constraint is beneficial to the 3D representation. Therefore, GAP3D is more like GaussianFormer rearranged in a grid manner rather than a novel planar representation.
3. Insufficient experiment. The paper only conducts experiments on Occ3D-nuScenes, which is quite limited.

**Questions:**

1. Since each primitive still adopts a Gaussian kernel formulation, I think they are actually Gaussian primitives as in GaussianFormer. Then the primitives of the same plane actually have little relationship with each other, except that the paper arranges them in a HxWxD format. What is the insight of this rearrangement?
2. How does the proposed method perform on other widely used benchmark, such as SurroundOcc?

---

> ### Author Response · Authors · 2025-11-23
>
> We thank the reviewer for the thoughtful comments and for noting the motivation, clarity and strong performance of GAP3D. We respond to the addressed concerns below.
> - **[W1] Related work.** We will add TPVFormer, GaussianFormer, and other relevant works that have come to our attention in the revised version. We also note that, although these methods appear closely related at first glance, their underlying representations differ substantially from GAP3D conceptually.
> - **[W2, Q1] Novelty.** We clarify the differences between GAP3D and GaussianFormer [2]. We believe that the statement “*GAP3D is GaussianFormer rearranged into a grid*” is an oversimplification that overlooks the core contributions and conceptual differences of our method. We highlight the key distinctions below:
>   - GaussianFormer represents the scene using 3D Gaussian primitives with semantic logits as one of its attributes. GAP3D does not use Gaussian primitives; it consists of adaptive 2D planes allocated using a stick-breaking formulation.
>   - GaussianFormer directly adopts the differentiable rendering function with visibility-aware transmittance used in Gaussian splatting [1], where nearby Gaussians partially occlude contribution from further ones. GAP3D does not use a rendering function; it merely employs Gaussian kernel function solely to propagate geometry inferred by the adaptive planes and ensure consistency between plane heights and voxel features.
>   - GaussianFormer’s allocates a fixed number of Gaussian primitives which are unconstrained within the occupancy prediction space. As in other sparse methods, primitives may concentrate in high-confidence regions, leaving weaker regions prone to missed obstacles. GAP3D avoids this issue by maintaining through its compact adaptive planar representations.
> - **[Q1] Intuition behind GAP3D’s design.** The adaptive planes are the core mechanism that gives GAP3D global coverage while still allocating representational capacity based on the underlying scene geometry. Through the stick-breaking process, the model performs causal, bottom-up reasoning conditioned on global context of previously allocated planes. This yields a geometry-aligned vertical decomposition with concentrated resolution near meaningful structures. The Gaussian kernel simply enforces geometric consistency between the learned plane heights and the resulting voxel features. Together, these components produce a structured, geometry-aware representation that is fundamentally different from Gaussian primitive-based methods and cannot be obtained by simply rearranging such primitives into a grid.
> - **[W3, Q2] Experiments.** We respectfully note that Occ3D-nuScenes is currently the most widely adopted and representative benchmark for multi-view semantic occupancy prediction in driving scenarios, and many recent works have reported results exclusively on this dataset. Regardless, we provide additional results on SurroundOcc and OpenOccupancy (please refer to our global response).
>
> [1] “3D Gaussian Splatting for Real-Time Radiance Field Rendering”, SIGGRAPH 2023\
> [2] “GaussianFormer: Scene as Gaussians for Vision-Based 3D Semantic Occupancy Prediction”, ECCV 2024

---

### Official Review · Reviewer_Nxvo · 2025-10-30

**Soundness:** 3
**Presentation:** 3
**Contribution:** 3
**Rating:** 4
**Confidence:** 4

**Summary:**

I offer a different perspective about the paper: This paper introduces a novel method where each layer predicts a relative update to the height of its query. The subsequent layer then utilizes this incremental adjustment to refine the query's location. Based on this updated position, features are extracted through an adaptive mixing mechanism. Crucially, by representing the data in a relative structured way, the model avoids the need for unordered signals like a CD loss, which significantly facilitate and stabilize the convergence process.

**Strengths:**

The paper is well organized and the logic is easy to follow.

The experiments achieve the SOTA performance.

**Weaknesses:**

1.	Since I consider the paper as a variant of the OPUS architecture, I have some questions: where does the overall speed advantage come from compared to the original OPUS model? How are the features for the initial plane in the first layer obtained?

2.	The RayIoU metric, while valuable, has a notable limitation: it only evaluates accuracy on non-free rays and fails to account for false positives predicted on free rays. Therefore, I believe RayIoU alone is insufficient for comprehensive model evaluation. In contrast, the mIoU metric could potentially capture this aspect of performance. For a more balanced and community-accessible benchmark, I recommend supplementing the results with the mIoU metric to provide a fuller picture of model capabilities.

3.	Given that modeling long-range dependencies is a key and unique aspect of the method, a specific ablation on this component is advisable. A more thorough description of this mechanism in the methodology would also improve clarity for readers.

4.	The experimental evaluation would benefit from additional visual comparisons with other methods to better demonstrate the qualitative advantages of the proposed approach.

5.	It is noted that SparseOcc, which also employs a dense representation, was trained for only 24 epochs, whereas the proposed method was trained for 60 epochs. Consequently, it remains unclear whether the observed performance gains stem from the novel representation or the substantially longer training schedule. A controlled ablation study to disentangle these two factors would significantly strengthen the paper's claims.

6.	The description of feature splatting needs to be explicitly stated: is it applied to all voxels in the 3D space or constrained to a single plane?

7.	The training strategy employs several unconventional design choices that require further justification. For instance, the rationale behind annealing the regularization loss weight to zero over the first six epochs, as well as the decision to change the batch size to 32 in the second training stage, should be explicitly explained.

**Questions:**

See weakness.

My final score is directly contingent upon the quality of the rebuttal. Should all of the raised concerns be adequately addressed, I will elevate the rating to an 8. Conversely, if the response fails to fully resolve any issues, the score will likely remain at its current level. However, a minor improvement to a 6 is also be considered if some of the concerns are satisfactorily clarified.

---

> ### Author Response · Authors · 2025-11-25
>
> We thank the reviewer for the thoughtful, careful reading of our work, and for providing a thorough perspective on our method. We address the reviewer's concerns in detail below.
> - **[W1] Comparison with OPUS.** We respectfully note that GAP3D is not a variant of OPUS [3]; the only shared aspect is that both are query-based occupancy prediction methods (similar to SparseOcc [1]), unlike LSS-based [2] methods which are most prevalent in recent works. We believe that the speed difference arises from fundamentally different design choices:
>   - OPUS is an unstructured sparse representation based method, where each query must predict not only the semantic class, but also precise 3D locations
>   - Due to this design, OPUS requires a coarse-to-fine refinement pipeline where each query predicts an increasing number of refinement points across transformer decoder layers (for OPUS-L, {1, 2, 4, 8, 16, 16}). The number of predicted 3D points per query grows substantially, which increases latency.
>   - GAP3D, in contrast, provides global scene coverage using a structured plane-based representation, so it does not require predicting large sets of 3D locations.
>   - On top of that, GAP3D uses Mamba layers instead of quadratic self-attention, enabling global receptive fields across a much large number of queries (40k, 10k, 2.5k), which would be infeasible with full attention.
>
>    We provide a table with architectural details to illustrate the difference:
>    | **Model** | **# Queries** | **Attention Complexity** | **# Sampling Points** | **# Predicted Points** |
>    | --- | --- | --- | --- | --- |
>    | **OPUS-L** | 28,800 (4,800×6) | O(Q²) ≈ 2.3×10⁷ | 115,200 (4 per query) | 225,600 (1,2,4,8,16,16 per query) |
>    | **GAP3D** | 105,000 (40k×2 + 10k×2 + 2.5k×2) | O(Q) ≈ 4×10⁴ | 140,000 (1,2,4 per query) | 105,000 (1 per query) |
>
>    **Initial plane features.** The first plane in the first layer is initialized solely through positional embeddings, so the initial heights effectively act as a learned bias (as in SparseOcc [1]). All subsequent planes are then predicted causally via the stick-breaking process.
> - **[W2, W4] Evaluation metrics and qualitative comparisons.** We agree with the reviewer’s assessment regarding the limitation of the conventional RayIoU metric. To address this, we redefine the original RayIoU as **RayIoU-Recall**, and introduce two complementary metrics that explicitly account for false positives.
>   - **RayIoU-Recall** (traditional RayIoU): Computed on rays that intersect *ground-truth* occupied voxels.
>   - **RayIoU-Precision**: Computed on rays that intersect *predicted* occupied voxels.
>   - **RayIoU-Union**: Computed on rays that intersect occupied voxels in either predictions or ground-truth.
>
>    | **Method** | **mIoU** | **RayIoU-Recall (1m, 2m, 4m)** | **RayIoU-Precision (1m, 2m, 4m)** | **RayIoU-Union (1m, 2m, 4m)** |
>    | --- | --- | --- | --- | --- |
>    | OPUS-L | **36.1** | 41.1 (34.5, 41.9, 46.7) | 40.6 (34.2, 41.4, 46.1) | 39.3 (33.1, 40.1, 44.6) |
>    | GAP3D | 35.3 | **42.8 (37.8, 43.6, 47.0)** | **46.3 (40.7, 47.3, 50.9)** | **41.8 (37.0, 42.7, 45.8)** |
>
>    GAP3D achieves higher RayIoU-Recall | Precision | Union across all depth thresholds, despite having slightly lower mIoU. This indicates that GAP3D produces fewer false positives while better capturing true occupied regions. We provide additional qualitative comparisons (Appendix A.5) that show that OPUS-L’s predictions tends to over-fill empty regions (*a behavior that mIoU does not penalize, as the reviewer is fully aware from prior work [1]*), leading to inflated mIoU but worse geometric fidelity. These results collectively strengthen our belief that RayIoU-based metrics provide a more reliable assessment of occupancy quality over mIoU.
>
>    (Note: The OPUS-L checkpoint provided by their repository yields slightly worse scores than those reported on their paper; the authors note that they provide only a re-trained version.)
> - **[W5] Training schedule comparisons.** We respectfully point out that SparseOcc reports 60-epoch results on their public repository, with only marginal improvement over its 24-epoch version (36.8 → 37.7 RayIoU). Furthermore, our 12-epoch version (Tab.2) already outperforms SparseOcc’s 24-epoch version on RayIoU (39.3 vs. 36.8). We also note that OPUS-L trains for 100 epochs. Regardless, we believe that the final converged performance is the most meaningful way to evaluate a representation’s capability.
>
>    | Method | Training Epochs | RayIoU (Avg) |
>    | --- | --- | --- |
>    | SparseOcc | 24 | 36.8 |
>    | SparseOcc | 60 | 37.7 |
>    | OPUS-L | 100 | 41.2 |
>    | GAP3D (ours) | 12 | 39.3 |
>    | GAP3D (ours) | 60 | 42.8 |
>
>    That being said, if time permits during the rebuttal period, we will provide additional ablations of GAP3D trained for different numbers of epochs.

---

> ### Author Response · Authors · 2025-11-25
>
> - **[W6] Clarification on geometry-aware splatting.** We appreciate the request for clarification. The geometry-aware splatting operation is applied to all voxels in the 3D occupancy space: each adaptive plane contributes to all voxels. We will provide a clearer description in the revised draft.
> - **[W7] Unconventional training details.** The design choices mentioned were primarily driven by practical considerations and resource constraints. We will clarify these details in the revised draft.
>   - The temporary use of a smaller batch size during initial training was simply a practical measure to quickly validate architectural variations. Once the best configuration was identified, we increased the batch size to fully utilize GPU capacity and complete experiments quickly.
>   - The height regularization loss is intended to guide the placement of adaptive planes in initial training. The height targets obtained for this regularization loss are not expected to be optimal and may hinder performance if applied throughout full training. For this reason, we chose to gradually anneal its weight, allowing the model to rely solely on voxel supervision to learn the optimal adaptive plane placements.
>
> [1] “Fully Sparse 3D Panoptic Occupancy Prediction”, ECCV 2024.\
> [2] “Lift, Splat, Shoot: Encoding Images from Arbitrary Camera Rigs by Implicitly Unprojecting to 3D”, ECCV 2020.\
> [3] “OPUS: Occupancy Prediction Using a Sparse Set”, NeurIPS 2024.\
> [4] “VMamba: Visual State Space Model”, NeurIPS 2024.

---

> ### Comment · Reviewer_Nxvo · 2025-11-27
> **Comments  by Reviewer**
>
> I think the authors have addressed most of my concerns, I thus raise the rating

---

> ### Author Response · Authors · 2025-11-28
>
> We sincerely appreciate the reviewer's updated assessment. As requested, we have finalized the ablation regarding the effect of long-range dependency reasoning.
> - **[W3] Ablation on long-range dependencies.** GAP3D uses Mamba layers for long-range reasoning. To isolate their effect, we remove Mamba layers and keep only the MLP layers.
>
>    | Method | RayIoU (%) | RayIoU@1m (%) | RayIoU@2m (%) | RayIoU@4m (%) | FPS |
>    | --- | --- | --- | --- | --- | --- |
>    | **GAP3D w/o Mamba** | 37.2 | 31.2 | 38.1 | 42.5 | 11.4 |
>    | **GAP3D** | 38.7 | 32.8 | 39.6 | 43.6 | 9.5 |
>
>    We have also attempted to replace Mamba layers with self-attention layers but found it to be computationally infeasible. We will add a clearer explanation of this mechanism in the revised version.

---

### Official Review · Reviewer_dNop · 2025-11-01

**Soundness:** 2
**Presentation:** 2
**Contribution:** 2
**Rating:** 4
**Confidence:** 5

**Summary:**

This paper introduces GAP3D, a framework for 3D occupancy prediction from multi-view images. The method aims to solve the trade-off between traditional dense voxel grids, which offer full coverage but scale poorly in memory and computation, and sparse representations, which are efficient but risk missing critical regions and lack global consistency. GAP3D combines the strengths of both approaches with two key contributions:
1. Adaptive Plane Representations: The model represents the 3D scene using a set of adaptive 2D planes. It employs a stick-breaking process to sequentially partition the vertical space, allowing the height of each plane to dynamically adapt to the scene's geometry. This focuses representational capacity on geometrically significant regions.
2. Geometry-Aware Splatting: These adaptive plane features are fused into a dense 3D occupancy volume using a differentiable Gaussian kernel. This operation preserves the learned geometric structure and produces spatially consistent predictions.

To aid training, the authors also introduce a height regularization loss ($\mathcal{L}_{height}$) to encourage the adaptive planes to align with the scene structure, especially in early training stages.
Experiments on the Occ3D-nuScenes benchmark demonstrate that GAP3D achieves state-of-the-art performance, attaining 42.8% RayIoU. It outperforms previous methods like STCOcc and OPUS while maintaining competitive efficiency.

**Strengths:**

1. The idea of using planes is not new (e.g., TPV ), but the method for their placement is highly original. The use of a stick-breaking process to sequentially and adaptively partition the vertical space is a clever mechanism. This allows the representation to dynamically allocate "representational capacity" to geometrically complex areas, unlike fixed grids.
2. The paper introduces a differentiable method to fuse these adaptively-placed 2D planes back into a 3D volume6666. Using a Gaussian kernel that is aware of the adaptively predicted plane heights ($h_{u,v}^{(i)}$) ensures that the geometric structure learned by the planes is coherently preserved in the final voxel output.
3. The results are state-of-the-art. GAP3D achieves the highest RayIoU (42.8%) on the benchmark, outperforming prior work like STCOcc. This SOTA performance is particularly impressive because it is achieved using fewer input frames (8 vs. 16) and a lower input resolution than the next-best method.

**Weaknesses:**

1. The proposed method using planes (TPVFormer) and Gaussian splatting (GaussianFormer) is not new.
2. The authors could run experiments on other benchmarks, such as nuScenes-OpenOccupancy and SemanticKITTI, to demonstrate the generalization ability of the proposed method.
3. It's unclear what the model learns if the scene lacks strong geometric features in a specific (u,v) column (e.g., open sky). The stick-breaking process still has to allocate $K$ planes.
4. The former sparse methods, like SparseOcc (both ECCV 2025 and CVPR 2024 versions), GaussianFormer (v1&v2), which use free x-y/u-v coordinate sparse voxels/3D Gaussians. This paper tries to fix the x-y coordinates to form planes. This may leed to the resource waste on empty areas.
5. The ablation study on the number of queries should be provided.
6. Table 3 explores the number of planes, but it does so with a reduced channel dimension (64 instead of 80) due to computational cost. This is a confounding variable. The performance drop at 6 planes (38.4 RayIoU) compared to 4 planes (38.7 RayIoU)  might not be because 6 planes are worse, but because 6 planes with fewer channels is worse. The conclusion that "simply adding more planes is not always advantageous"  is not fully supported by this experiment.
7. The speed of the proposed method seems to be much slower than the previous methods.

**Questions:**

1. Could you provide more intuition on the learned behavior of the adaptive planes in geometrically empty regions, such as the sky?
2. Could you clarify if the 4-plane result in this table also used 64 channels? If not, could you provide a "cleaner" ablation where the channel dimension is held constant to isolate the true impact of $K$?
3. The "geometry-aware splatting" via a Gaussian kernel is a key component. This method shares conceptual similarities with other well-established techniques (e.g., GaussianFormer). Could you elaborate on the specific novelty of your splatting mechanism?

---

> ### Author Response · Authors · 2025-11-23
>
> We thank the reviewer for the thoughtful feedback, especially for highlighting the originality of the adaptive plane placement, and the strong performance of GAP3D. We address the weaknesses and questions raised by the reviewer below.
> - **[W1, Q3] Novelty.** We clarify that **GAP3D does not perform Gaussian splatting [1]** and is fundamentally different from methods that adopt it (e.g., GaussianFormer [3]). In Gaussian splatting, the scene is modeled directly as a set of Gaussian primitives whose attributes are rendered through a visibility-aware transmittance function, where closer Gaussians can partially occlude contributions from further ones. This rendering mechanism is central to 3D Gaussian splatting. In contrast, GAP3D neither uses Gaussian primitives nor the differentiable rendering function. The Gaussian kernel in GAP3D is simply a differentiable weighing function used to propagate the geometry predicted by the adaptive planes, ensuring that voxel features remain consistent with the learned plane heights and spreads. While this kernel itself is not novel, its role together with the adaptive, causally constructed planes is essential. This combination yields a 3D representation that is fundamentally different from GaussianFormer (which directly performs Gaussian splatting), and to the best of our knowledge has not been proposed before.
> - **[W1] Novelty.** Regarding TPVFormer [2], we believe the similarity is not straightforward. TPVFormer constructs three fixed, orthogonal planes in order to decompose the 3D scene voxel grid into top-front-side views for efficient processing. In contrast, GAP3D predicts a sequence of causally dependent, adaptively placed planes along the vertical range using a stick-breaking formulation. This yields a causally interdependent, geometry-aligned 3D representation that is conceptually different from TPVFormer’s tri-perspective view representation. We illustrate this difference in Fig. 2. We would appreciate clarification from the reviewer on which specific component of TPVFormer they find similar to GAP3D.
> - **[W2] Experiments.** Please refer to our global response, where we provide additional results on SurroundOcc and OpenOccupancy.
> - **[W3, Q1] Modeling empty regions.** In cases where $(u,v)$ column contains little or no geometry (e.g., open sky with only ground below), the model simply learns to represent empty free space, just as any other dense 3D representation would. It is worth noting that GAP3D allocates adaptive plane heights by using global scene context, so geometry-weak regions are still guided by surrounding cues rather than solely relying on that column. As shown in the qualitative results in Fig. 4, the stick-breaking process tends to place plane heights similar to those of nearby structures.
> - **[W4] Efficiency comparison with sparse methods.** We note that sparse methods also model empty regions in practice, and therefore still allocate computation to non-ignorable amounts of free space, just as GAP3D does. As shown by the FPS results in Tab. 1, it is not evident that sparse methods typically waste fewer resources on empty areas: OPUS [4], the current state-of-the-art sparse method, achieves lower FPS than GAP3D (7.2 vs 9.0), and GaussianFormer [3] reports a latency of 372 ms (2.7 FPS) on a RTX 4090 GPU (Tab. 3 [3]) for producing an occupancy volume of the same resolution (200$\times$200$\times$16). On top of that, sparse methods may miss crucial obstacles due to fixed voxel budgets or pruning heuristics, whereas GAP3D’s compact representation maintains global coverage of the full scene.
> - **[W5] Ablation on queries.** Query count is tied to the number of planes, which we already ablate in Table 3. If the reviewer has a different ablation in mind, we welcome any clarifications.
> - **[W6, Q2] Clarification on results.** This is a misunderstanding. All results in Table 3 were obtained with the same channel dimension of 64. The larger channel dimension of 80 was used only for the main results (Table 1), and Table 2 when ablating key components. We will make this distinction clearer in the revised version.
> - **[W7] Trade-off between accuracy and efficiency.** There is naturally a trade-off between performance and efficiency, and the goal is to achieve a practical balance. While some earlier methods are faster (e.g., FlashOcc, SparseOcc, FBOcc), they also exhibit noticeably lower accuracy. For applications such as autonomous driving, where safety is of utmost importance, the improvement in prediction accuracy offered by GAP3D is substantial and, in our view, justifies the moderate increase in computation.
>
> [1] “3D Gaussian Splatting for Real-Time Radiance Field Rendering”, SIGGRAPH 2023\
> [2] “Tri-Perspective View for Vision-Based 3D Semantic Occupancy Prediction”, CVPR 2023\
> [3] “GaussianFormer: Scene as Gaussians for Vision-Based 3D Semantic Occupancy Prediction”, ECCV 2024\
> [4] “OPUS: Occupancy Prediction Using a Sparse Set”, NeurIPS 2024

---

### Official Review · Reviewer_W6uu · 2025-11-01

**Soundness:** 3
**Presentation:** 3
**Contribution:** 2
**Rating:** 4
**Confidence:** 3

**Summary:**

This paper proposes a novel scene representation approach for the task of 3D semantic occupancy prediction. The core idea is to elevate the traditional BEV (Bird's Eye View) grid into spatial planes, and aggregate these plane features into voxel features using geometry-aware splatting within 3D space for subsequent occupancy prediction. Experiments conducted on the Occ3D-nuScenes benchmark demonstrate the effectiveness of the proposed method.

**Strengths:**

Originality：
GAP3D introduces a novel approach by lifting BEV features into 3D planes, which are then aggregated into voxel features via splatting. This representation paradigm enhances the spatial expressiveness of BEV features to some extent.

Quality：
The method is well-motivated and mathematically well-defined.

Clarity：
The presentation is clear and well-structured. Figures progressively illustrate the proposed representation and pipeline, notations are consistent, and the appendix provides helpful implementation details and information on numerical stability.

Significance：

This work proposes an improved representation paradigm for occupancy prediction in autonomous driving, and demonstrates noticeable performance gains. It can serve as a useful reference for future research in this area.

**Weaknesses:**

Novelty：
Although the paper claims to introduce a new plane-based representation, it essentially amounts to enhancing BEV features along the height dimension. Similar ideas have already been explored in the Pillar-based literature. Thus, the proposed representation does not appear sufficiently novel.

Experiments：
The paper only reports results on the Occ3D-nuScenes dataset. However, other occupancy prediction datasets such as Occ3D-Waymo and SurroundOcc, have been widely adopted. The current experimental evaluation is therefore insufficient.

**Questions:**

1. Limited Dataset Comparisons: As far as I know, there are several autonomous driving datasets for occupancy prediction beyond nuScenes, such as Waymo and KITTI. Additionally, multiple benchmarks exist besides Occ3D. The paper only evaluates on Occ3D-nuScenes and should include additional experiments for a more comprehensive comparison.

2. Evaluation Metrics: The paper only uses RayIoU for evaluation, but mIoU is a more widely used metric for occupancy prediction. This important metric is missing and should be included.

3. Plane Representation Concerns: The proposed plane representation merely adds one more dimension to the BEV grid and lifts it into 3D space. This can hardly be regarded as a truly sparse representation. In contrast, existing works like GaussianFormer have implemented  primitive-based occupancy prediction, which this paper fails to mention.

---

> ### Author Response · Authors · 2025-11-23
>
> We thank the reviewer for the positive feedback, especially noting that GAP3D can serve as a useful reference for future work. Below, we address the reviewer’s concerns.
>
> **Regarding novelty:**
> - **GAP3D is fundamentally different from pillar-based BEV representations.** To make this distinction explicitly, we compare with BEVFormer [1], a representative pillar-based baseline. In BEVFormer, the reference points along the $z$-axis are (1) *predetermined*, (2) *evenly distributed*, and (3) completely *independent* of each other. In contrast, GAP3D uses a stick-breaking formulation to predict *causally dependent* reference points, enforcing bottom-up reasoning with adapative resolution along the vertical range. Furthermore, each adaptive plane is conditioned on *global scene context* before contributing to successive planes, allowing the model to reason over the entire scene rather than a single pillar query when predicting the next plane’s height. These architectural differences, together with our geometry-aware splatting mechanism, produce an adaptive and geometry-aware decomposition of space rather than a simple extension of BEV features along the vertical range. Empirically, results in Tab.2 further demonstrate the significance of this design. Regarding the statement that our representation “merely adds one more dimension to the BEV grid”, we believe this statement is an oversimplification that overlooks the key contributions of our method.
>
> - **We clarify that GAP3D is neither a sparse representation nor a primitive-based one.** We do not claim GAP3D is a sparse framework anywhere in the paper. Instead, we design a compact representation that retains some benefits of sparse methods while avoiding their failure modes, such as missing objects due to fixed number of voxel predictions or pruning heuristics. Likewise, GAP3D is not related to primitive-based methods such as GaussianFormer [2]. GaussianFormer represents the scene with 3D Gaussian primitives whose attributes include semantic logits, and the final occupancy field is obtained directly by rendering or splatting these primitives onto the voxel grid. In contrast, ***GAP3D does not use Gaussian primitives.*** We only adopt a Gaussian kernel as a differentiable way to propagate the geometry predicted by adaptive planes, ensuring that voxel features remain consistent with the learned plane heights and spreads. The Gaussian kernel in GAP3D is simply a smoothing and geometry-aware aggregation mechanism, not a primitive-based representation. Thus, even though both methods employ Gaussian-based functions, their underlying representations and inductive biases are fundamentally different.
>
> - In addition, we would appreciate any references if the reviewer has specific pillar-based methods that employ similar sequentially or globally conditioned reasoning, as we are not aware of such prior work. We also thank the reviewer for pointing out missing related works like GaussianFormer [2] and will include it in the revised draft.
>
> **Regarding experiments:**
> - Please refer to our global response, where we provided mIoU results on SurroundOcc and OpenOccupancy. Regarding Occ3D-Waymo and SemanticKITTI specifically, we did not include them in the initial submission because recent 3D occupancy works rarely report results on these datasets, making comparisons difficult. Nonetheless, we intend to report additional results if we are able to complete these experiments within the rebuttal period.
>
> [1] “BEVFormer: Learning Bird’s-Eye-View Representation from Multi-Camera Images via Spatiotemporal Transformers”, ECCV 2022.\
> [2] “GaussianFormer: Scene as Gaussians for Vision-Based 3D Semantic Occupancy Prediction”, ECCV 2024.

---

> ### Author Response · Authors · 2025-11-25
>
> **Regarding evaluation metrics:**
> - **mIoU is known to be unreliable on Occ3D-nuScenes [5].** SparseOcc [3] explicitly points out that mIoU on Occ3D does not penalize models that over-fill empty regions, because large portions of free space are masked out during evaluation (visualized as grey voxels in Appendix A.5; note that for clarity we exclude masked empty annotations in these figures). Our qualitative comparisons show this effect clearly: OPUS-L [4], despite achieving higher mIoU than GAP3D, exhibits non-negligible overfilling behaviour in occluded regions.
> - **Addressing RayIoU’s limitation.** As reviewer Nxvo noted, the traditional RayIoU metric introduced in [3] does not account for false positive rays. To provide a more complete evaluation, we introduce two additional RayIoU variants:
>     - **RayIoU-Recall** (traditional RayIoU): evaluate rays where ground-truth is occupied.
>     - **RayIoU-Precision**: evaluate rays where model predictions are occupied, thus accounting for false positives.
>     - **RayIoU-Union**: evaluate rays where either ground-truth or model predictions are occupied, serving as a more balanced metric.
>
>     We compare results using such metrics against OPUS-L [4].
>
>     | **Method** | **mIoU** | **RayIoU-Recall (1m, 2m, 4m)** | **RayIoU-Precision (1m, 2m, 4m)** | **RayIoU-Union (1m, 2m, 4m)** |
>     | --- | --- | --- | --- | --- |
>     | OPUS-L | **36.1** | 41.1 (34.5, 41.9, 46.7) | 40.6 (34.2, 41.4, 46.1) | 39.3 (33.1, 40.1, 44.6) |
>     | GAP3D | 35.3 | **42.8 (37.8, 43.6, 47.0)** | **46.3 (40.7, 47.3, 50.9)** | **41.8 (37.0, 42.7, 45.8)** |
>
>     GAP3D consistently achieves higher RayIoU-Recall | Precision | Union across all depth thresholds, indicating fewer false positives and fewer false negatives, despite slightly lower mIoU. We will add these metrics in the revised draft.
>
> [3] “Fully Sparse 3D Panoptic Occupancy Prediction”, ECCV 2024.\
> [4] “OPUS: Occupancy Prediction Using a Sparse Set”, NeurIPS 2024.\
> [5] “Occ3D: A Large-Scale 3D Occupancy Prediction Benchmark for Autonomous Driving”, NeurIPS 2023.

---

### Author Response · Authors · 2025-11-21
**General Response: Additional Experiments**

We sincerely thank all the reviewers for their constructive feedback and for recognizing the motivation (W6uu, uiDp), originality (W6uu, dNop), writing clarity (W6uu, dNop, uiDp), and experiment results (dNop, Nxvo, uiDp) of our work. A common concern raised across reviewers is the limited variety of benchmark evaluations. In response, we have conducted additional experiments and report the results below. We also intend to evaluate our method on OpenOcc [3] and Occ3D-Waymo [4] and will add results when completed.

**Experiments on SurroundOcc [1].** We evaluate GAP3D on the SurroundOcc benchmark, using reference numbers reported in Inverse++ [5]. For a fair comparison, we match the best performing model’s training epochs (24 epochs). Despite using significantly fewer parameters and a smaller backbone, GAP3D outperforms Inverse++ by 3.1 mIoU. We appreciate the reviewers’ suggestions and will include these results in the revised draft.

| Method           | Backbone | Params | IoU (%) | mIoU (%) |
|------------------|----------|--------|---------|----------|
| BEVFormer        | R101     | 59M    | 30.50   | 16.75    |
| TPVFormer        | R101     | 69M    | 30.86   | 17.10    |
| OccFormer        | R101     | 169M   | 31.39   | 19.03    |
| FB-Occ           | R101     | -     | 31.50   | 19.60    |
| SurroundOcc      | R101     | 180M   | 31.49   | 20.30    |
| GaussianFormer   | R101     | -     | 29.83   | 19.10    |
| GaussianFormer-2 | R101     | 125M   | 31.74   | 20.82    |
| Inverse++        | R101     | 137M   | 31.73   | 20.91    |
| **GAP3D**        | R50      | **78M**| **36.64** | **24.00** |

**Experiments on OpenOccupancy [2].** OpenOccupancy uses a large voxel resolution (512x512x40), so following SparseOcc [6], we obtain features at a reduced size (128x128x10) and then upsample these to full resolution occupancy via interpolation and a lightweight linear head. We report results from SparseOcc [6], and observe that GAP3D outperforms it by 2.1 mIoU, even though our model was not specifically designed to produce such high-resolution occupancy outputs. We will provide details and tables in the revised draft.

| Method | Backbone | IoU (%) | mIoU (%) |
| --- | --- | --- | --- |
| MonoScene | R50 | 18.4 | 6.9 |
| TPVFormer | R50 | 15.3 | 7.8 |
| OpenOccupancy | R50 | 19.3 | 10.3 |
| C-CONet | R50 | 20.1 | 12.8 |
| SparseOcc | R50 | 21.8 | 14.1 |
| **GAP3D** | R50 | **28.7** | **16.2** |

[1] “SurroundOcc: Multi-Camera 3D Occupancy Prediction for Autonomous Driving”, ICCV 2023\
[2] “OpenOccupancy: A Large Scale Benchmark for Surrounding Semantic Occupancy Perception”, ICCV 2023\
[3] “Scene as Occupancy”, ICCV 2023\
[4] “Occ3D: A Large-Scale 3D Occupancy Prediction Benchmark for Autonomous Driving“, NeurIPS 2023\
[5] “Inverse++: Vision-Centric 3D Semantic Occupancy Prediction Assisted with 3D Object Detection”, arXiv 2025\
[6] “SparseOcc: Rethinking Sparse Latent Representation for Vision-Based Semantic Occupancy Prediction”, CVPR 2024

---

### Author Response · Authors · 2025-12-02
**Summary of Rebuttal and Discussion**

We would like to provide a brief summary of our rebuttal and discussion given the extraordinary circumstances. We responded in detail to all reviewer concerns and clarified several misunderstandings regarding GAP3D's novelty, design, experiment setup and results. Notably, we have:
- Performed additional experiments on **SurroundOcc** [1] and **OpenOccupancy** [2].
- Provided a detailed explanation of the limitations of **mIoU** on Occ3D-nuScenes (as also noted in prior work [3]).
- Introduced **RayIoU-Recall, RayIoU-Precision, and RayIoU-Union** to address the limitations of the current RayIoU metric [3] (evaluation only on non-free rays, as noted by reviewer Nxvo), and showed that GAP3D performs strongly under these more balanced metrics.

During the review period, one of the reviewers explicitly stated that their concerns had been resolved and raised their score, while other reviewers did not have the opportunity to respond before the discussion was closed.
We hope this summary assists in reviewing the key points raised in the discussion.
We are happy to provide any additional clarification if needed.

[1] “SurroundOcc: Multi-Camera 3D Occupancy Prediction for Autonomous Driving”, ICCV 2023.\
[2] “OpenOccupancy: A Large Scale Benchmark for Surrounding Semantic Occupancy Perception”, ICCV 2023.\
[3] “Fully Sparse 3D Panoptic Occupancy Prediction”, ECCV 2024.

---

### Meta-Review · Area_Chair_Nn4D · 2026-01-04

**Summary:**

For the initial submission, reviewers have common concerns regarding experiments, including only reporting results on the Occ3D-nuScenes dataset, and only using RayIoU for evaluation, while mIoU is a more widely used metric for occupancy prediction and there are many other dataset such as SurroundOcc, Occ3D-Waymo, SemanticKITTI and OpenOccupancy. During the rebuttal, the authors provide additional results on OpenOccupancy and SurroundOcc with mIoU.

Another major concern is that three out of the four reivewers are concerned about the novely, and the initial submission does not cite closely related work such as TPVFormer, GaussianFormer, maiking the positioning of the manuscript difficult for readers. During the rebuttal, the authors emphasized the conceptual difference against GaussianFormer and TPVFormer, to clarify the novelty and contribution of the proposed method, more detailed ablation study and visual comparison with GaussianFormer and TPVFormer are necessary, which requires substantial changes and another round of review.

**Reviewer Concerns:**

Solved:
Lack experiments on SurroundOcc.
The paper only uses RayIoU for evaluation, but mIoU is a more widely used metric for occupancy prediction.
Many other detailed queries regarding clarity.
Comparison with OPUS.
The impact of training epochs.
Need more qualitative comparisons.
Lack of related work. The paper does not cite closely related work such as TPVFormer, GaussianFormer, etc.


Still outstanding concerns:
Novelty is limited (W6uu, dNop, uiDp), especially when positioned regarding TPVFormer and GaussianFormer.
Lack experiments on Occ3D-Waymo, SemanticKITTI (W6uu, dNop).
Lack visual comparisons with other methods (Nxvo). Only OPUS-L  is compared during rebuttal.

**Reviewer Scores:**

W6uu might keep 4 because of the concern on novelty.
dNop might keep 4 because of the concern on novelty.
Nxvo might raise from 4 to 6 as he thought the authors addressed most of his concerns.
uiDp might keep 4 beacause novelty.

---

### Decision · Program_Chairs · 2026-01-26

Reject